# Diversify and Conquer: The Vaccine Escapism of *Plasmodium falciparum*

**DOI:** 10.3390/microorganisms8111748

**Published:** 2020-11-07

**Authors:** Alena Pance

**Affiliations:** The Wellcome Sanger Institute, Genome Campus, Hinxton, Cambridgeshire CB10 1SA, UK; alena.pance@sanger.ac.uk

**Keywords:** *Plasmodium falciparum*, malaria, vaccine, variation, diversity, genomics, sequencing

## Abstract

Over the last century, a great deal of effort and resources have been poured into the development of vaccines to protect against malaria, particularly targeting the most widely spread and deadly species of the human-infecting parasites: *Plasmodium falciparum*. Many of the known proteins the parasite uses to invade human cells have been tested as vaccine candidates. However, precisely because of the importance and immune visibility of these proteins, they tend to be very diverse, and in many cases redundant, which limits their efficacy in vaccine development. With the advent of genomics and constantly improving sequencing technologies, an increasingly clear picture is emerging of the vast genomic diversity of parasites from different geographic areas. This diversity is distributed throughout the genome and includes most of the vaccine candidates tested so far, playing an important role in the low efficacy achieved. Genomics is a powerful tool to search for genes that comply with the most desirable attributes of vaccine targets, allowing us to evaluate function, immunogenicity and also diversity in the worldwide parasite populations. Even predicting how this diversity might evolve and spread in the future becomes possible, and can inform novel vaccine efforts.

## 1. Introduction

Although preventable and curable, malaria is one of the most severe worldwide public health problems, causing crippling disease leading to approximately half a million deaths every year. The intense efforts for malaria intervention deployed so far have achieved a substantial decrease in the incidence of the disease, which has encouraged ambitious plans for malaria elimination by the World Health Organisation. However, recent years have witnessed a stagnation of progress in the reduction of cases [1], compounded by a number of factors such as conflict, poverty and political instability.

Malaria is a complex infectious disease caused by five different species of *Plasmodium* parasites. *Plasmodium vivax* and *Plasmodium falciparum* are the most common, though their distribution around the world does not completely overlap. *Plasmodium vivax* and *Plasmodium ovale* form hypnozoites, latent hepatic forms that are difficult to clear and constitute a reservoir of parasites that maintains the infection in the population. *Plasmodium knowlesi* has a shorter life cycle and as a consequence a very quick onset of clinical symptoms. The most severe disease is caused by *Plasmodium falciparum*, and much effort has been devoted over decades of research to the development of a vaccine specific to this species [2]. All stages of the parasite in the human host have been targeted (Figure 1) with the aim of designing vaccines that prevent infection by directing the immune response to the sporozoites, that avoid clinical symptoms and the spread of disease by using the antigens expressed during the blood cycle, or that stop transmission by targeting gametocytes.

In this genomic era, the completion of the whole sequence of the *Plasmodium falciparum* genome [3] has revealed the full set of genes, paving the way to start deciphering their function in the different stages of the life cycle of the parasite. This knowledge has enhanced the understanding of the disease, and started providing a genomic landscape of the parasite in different regions of the world [4]. Being able to sequence parasite genomes from vast numbers of clinical samples allows for associations with characteristics such as drug resistance as well as an overall view of its distribution and spread [5]. Parasite evolution, population distribution and dynamics in different regions of the world [6,7,8] can also be estimated with genomic information. Importantly, these studies are revealing the complexity of parasite diversity, providing a detailed insight into the genetic variation of parasite proteins that are at the centre of vaccine development.

This review summarises the main vaccine developments for *Plasmodium falciparum*, particularly focusing on those that have reached clinical trials, providing efficacy data in humans [2] (Table 1). The results of these efforts are considered in light of the vast sequencing data of *P. falciparum* acquired over the last decade to discuss the current understanding of genomic variation and its impact on vaccine efficacy and development.

## 2. The Infection: The Sporozoite

Human malaria infections start when the sporozoite form of the parasite is deposited in the skin by the infected mosquito during its blood meal. The sporozoites move randomly in the skin until they contact the endothelium of lymphatic or blood circulation, which can take between 1 and 3 h. Importantly, during this time they are exposed to the immune system, particularly if they enter the lymph. Sporozoites have to traverse epithelial cells and cross the endothelial barrier to access the blood vessel, which involves the sporozoite microneme protein essential for cell traversal SPECT1 and 2 [41], cell traversal protein for ookinetes and sporozoites CelTOS [42] and a phospholipase (PL) [43,44]. Once in the circulation, the sporozoites are rapidly carried to the liver, where they are arrested by binding to the highly sulphated heparan sulphate proteoglycans (HSPGs) of the liver sinusoid through the circumsporozoite protein (CSP), which densely covers the surface of the sporozoites [45]. Access to the sinusoid is facilitated by the fenestrated endothelium, which allows the sporozoites to leave the blood stream migrating through or between endothelial cells. Upon contact with the hepatocytes, motility becomes essential for invasion and the thrombospondin-related anonymous protein (TRAP) is a key component of the actinomyosin motor of the sporozoite [46,47]. After migration around several hepatocytes, the sporozoite adheres to and penetrates the target cell in a process involving TRAP, CSP and apical membrane antigen 1 (AMA-1), establishing itself within a parasitophorous vacuole (PV) [48,49], where the transformation into the merozoites that will initiate the blood cycle takes place [50,51] (Figure 2).

Targeting this stage for vaccine development has several advantages. While *Plasmodium* is an obligate intracellular parasite, during this stage it is free in the blood stream for several hours and therefore accessible to the immune system. Furthermore, as much as 20% of the inoculum enters the lymph, where exposure to the immune system and stimulation of T lymphocytes can be very effective [52,53]. Many of the unsuccessful sporozoites are removed from the skin by antigen presenting cells such as dendritic cells, providing additional immune stimulation opportunities. Together with the low number of sporozoites inoculated, this could make an immune response very effective at eliminating the parasite. Targeting this stage would prevent infection all together, avoiding clinical symptoms and blocking transmission by eliminating the human reservoir of the parasite [54]. Many of the known pre-erythrocytic proteins important for hepatocyte invasion and survival in the liver have been exploited for vaccine development [55].

The key protein for the motility and invasion of hepatocytes as well as mosquito salivary glands, **TRAP**, is expressed in the micronemes and on the surface of sporozoites. Numerous studies have described TRAP as a target of cellular and humoral immunity, and this response is correlated with a reduced risk of clinical malaria [56] and sterile protective immunity [57]. A multi-epitope vaccine was constructed with the aim of inducing a broader immune response more representative of the parasite.

The most extensively tested of these vaccines is **ME-TRAP**. This is a multiepitope string that includes CD8 and CD4 T-cell epitopes from CSP, EXP-1 (exported protein 1), LSA (liver stage antigen) 1 and 3 and STARP (sporozoite threonine and aparagine-rich protein) fused to full length TRAP from the *P. falciparum* strain T9/96 from Thailand. All these proteins have been demonstrated to induce the immune system, and some have been trialled individually without great success [58]. Trials with **ME-TRAP**, assessing efficacy with a CHMI using the heterologous strain 3D7, showed 10% to 33% partial protection and 10% to 20% sterile protection [9,10,11,59]. Natural infections after vaccination of adult volunteers reduced the risk of infection by 67% [12]. Another study found an efficacy of 10% in Gambia [13], while in Kenyan children no significant difference between the control and vaccinated groups was observed [60]. Despite a non-significant efficacy in Senegal, the vaccine showed promise in Gambian and Burkinabe children, but when expanded to a hyperendemic area of Burkina Faso, no significant protective efficacy could be shown in children with previous exposure to malaria [14]. One of the reasons attributed to the failure of TRAP-based vaccines was the antigenic diversity of the protein. The vaccine TRAP comes from the Thai strain T9/96 [61] that differs by 6% from the amino acid sequence of the 3D7 strain, including major differences in the number of repeats of the PPN sequence. The extent of nucleotide variation in TRAP correlates with the transmission intensity in endemic areas. Accordingly, the highest diversity was found in African countries, particularly Gambia, Senegal and Uganda, where the potential for multiclonal infection is considerable. Furthermore, high polymorphisms were found in some of the B- and T-cell epitopes, whereas the functional domains tend to be more conserved [62]. Among the components of the ME string, LSA-1 was found to have significant non-synonymous variation [63], though none of these changes affected T-cell epitopes [64]. The comparison of STARP among isolates worldwide revealed 20 haplotypes of the coding sequence but low levels of codon changes, and some regions of the gene were perfectly conserved [65]. An overview of genome-wide variation in over 600 samples worldwide provided in Plasmoview [66] shows a high degree of variation in TRAP as well as EXP1, LSA-1 and LSA-3, though the latter shows some conservation in the C-terminus. STARP, on the other hand, seems fairly conserved, with most variation contained within two defined regions. The most extensive variation was observed in CSP with hardly any conserved areas, as described in detail below.

The major surface antigen of the sporozoite, **CSP**, has been known to induce antibodies that can confer protective immunity against malaria, and a mechanism of targeting the parasite in the skin stage has been proposed [67]. CSP has three domains: a conserved amino-terminal region, a central B-cell epitope containing 37–44 NANP amino acid repeat sequences, and a polymorphic carboxy-terminal region that elicits a T-cell response. This protein has been the centre of many attempts to design an effective vaccine. A range of versions were used in vaccine design, some minimal, containing a three-unit repeat peptide from the central region of the protein [68], combinations of epitopes [69] and long synthetic peptides [70], but the most extensively tested and the only licensed anti-malaria vaccine to date is RTS,S/AS01. It consists of 19 NANP repeats and the C-terminal region of CSP from the 3D7/NF54 *P. falciparum* lab strain, fused with the hepatitis B surface protein and reconstituted with a novel adjuvant AS01. A series of trials showing safety, immunogenicity and efficacy against infection led to a large-scale phase III trial involving over 15,000 children throughout Africa [15,16,17,18,19]. Clinical malaria was reduced by 28%, and a booster applied 18 months after the third dose increased efficacy to 36.3%. CHMI trials comparing the dose regimens of vaccination showed an improved efficacy ranging from 55% to 86% against infection with a homologous 3D7 parasite [20,21]. In order to assess whether the modest efficacy of the vaccine in protecting against natural infections is due to allele specificity, samples from 7000 vaccinated and non-vaccinated children were sequenced, revealing a reduction in vaccine efficacy from 50%, for parasites with a perfect match to the vaccine protein, to 33% when amino acid differences were present [71]. A wider comparison of samples from other continents found 393 unique PfCSP haplotypes, and only 5.3% of the sequences identified in Africa and 0.25% in Asia corresponded to the 3D7 haplotype [72]. The global worldwide frequency of the 3D7 haplotype in the vaccine CSP region was calculated at 1.71%. In the analysis, the N-terminus showed much lower diversity across continents [73], which can be also observed in PlasmoView [66]. Indeed, comparing variability in the three regions of CSP, the C-terminus polymorphisms are most complex in the African continent, and the central repeat region shows great diversity of haplotypes, including the number of repeats, which is higher in Oceania and Asia, while the N-terminus is the most conserved, amounting mainly to a 57 bp insertion very frequent in Asia, Oceania and South America [73]. The lower diversity in the N-terminal region of the protein, which is functionally important and plays a role in immunity, makes the inclusion of this region in an improved version of the vaccine an attractive possibility [74]. A combination of ME-TRAP and RTS,S has also been tested in a CHMI with homologous parasites, but no improvement of the efficacy achieved with the best formulation of RTS,S alone was found [22].

**CelTOS**, one of the *P. falciparum* cell-traversal proteins, is present in the mosquito stage ookinetes as well as sporozoites. It was identified by naturally acquired antibodies in an endemic region as associated with protection against symptomatic infection [75]. A recombinant vaccine of CelTOS was tested in mouse models, in which it induced both humoral and cellular immune responses and sterile protection [76,77]. The *P. berghei*-expressing *P. falciparum* CelTOS tested in this model showed inhibition of both sporozoite infection and ookinete to oocyst development in the mosquito [78]. A recombinant vaccine using a 522 bp fragment from the 3D7 lab strain (PfCelTOS FMP012) was tested in two phase I clinical trials to monitor the elicited immune response and assess efficacy in controlled human malaria infection (CHMI), but the results have not been published. The genetic diversity of *P. falciparum* full length CelTOS evidenced 39 non-synonymous SNPs in 34 positions in isolates worldwide, in comparison with 3D7, amounting to 66 haplotypes [79]. Nevertheless, it was determined that most of the predicted B- and T-cell epitopes are in conserved regions of the protein, while most of the variation concentrates in the C-terminus. The assessment of the efficacy of this vaccine will prove very interesting, particularly in the context of heterologous infections.

## 3. Whole Sporozoite Vaccines

Immunisation with dead or inactivated whole organisms is the most widely used strategy to induce a preventive immune response against infectious agents. It was also applied to malaria as early as 1945 in studies with *P. knowlesi* in monkeys [80], *P. vivax* in humans [81] and *P. berghei* in mice [82]. These attempts showed that it is possible to use whole parasite immunisation for malaria, which could be a more effective way to elicit a wider and more diverse response than that achieved with individual proteins.

Three types of interventions have been tested: the injection of irradiation-attenuated parasites, genetically attenuated parasites and live parasites under drug pressure. The attenuated sporozoites arrest development in the liver, while the live parasites reach the blood but are wiped out by the drug before establishing a blood infection.

The most clinically developed *P. falciparum* sporozoite (PfSPZ) vaccine consists of metabolically active, irradiation-inactivated sporozoites from the lab strain NF54. Controlled infections proved that the vaccine can provide sterile immunity when volunteers are challenged with the homologous parasite [23,24,25]. The vaccine was subsequently tested for performance against naturally occurring infections in Mali [26] and Tanzania [27], where efficacy dropped to 29% and 20%, respectively. This was a clear indication that the lab strain NF54 vaccine is much less effective in protecting against heterologous parasites. A study was set up specifically to address this issue with a controlled infection with either homologous 3D7 parasites or the heterologous 7G8 from Brazil [28]. The results showed a 92% protection against 3D7 (12 of 13) after 3 weeks that dropped to 70% (7 of 10) after 24 weeks, while the protection against 7G8 was 80% (4 of 5) 3 weeks after immunization, which dropped to 10% (1 of 10) after 24 weeks. Though numbers in this study were small, there is an indication that the vaccine is less efficient against heterologous parasites and that immunity wanes off more rapidly with time.

In order to avoid potential damage of the SPZs by the irradiation process, which can have a negative impact on the vaccination efficacy, genetic inactivation was developed. These parasites can progress further through the liver stages and are therefore deemed more effective at triggering an immune response. Attenuation is achieved by deleting crucial genes involved in parasite development and, as a consequence, arrest in the liver stage [83]. The first clinical trial with this strategy was performed using parasites lacking p52 and p36, but there was a breakthrough leading to infection [84]. Genetic attenuation was improved by deleting p52, p36 and sap1, producing PfGAP3KO parasites that arrest early in the liver stage [85]. No efficacy results are available yet for these vaccines, but given that they originate from genetically engineered parasites, they are likely to rely on specific lab strains. It remains to be determined whether in this form, the stimulation of an immune response can be strong enough to be effective against a wider range of parasites.

Immunisation with live parasites under prophylaxis has been performed using mosquitoes infected with chloroquine-sensitive NF54 parasites to deliver parasites to volunteers, together with chloroquine treatment. Challenge with the homologous parasite showed 100% sterile immunity in the vaccinated participants, while the non-vaccinated individuals all developed the infection [24,29]. A comparison of the protection with this vaccination strategy against challenges with the genetically distinct clones NF135.C10 from Cambodia and NF166.C8 from Guinea was performed [30]. Consistent with the previous studies, vaccination provided 100% sterile immunity to the homologous parasite, but achieved 10 to 20% protection against the heterologous clones. Whole genome sequencing using various platforms was used to compare the vaccine strain NF54 with the clone 3D7, as well as strains used for heterologous challenges (7G8, NF166.C8 and NF135.C10), and also with a collection of clinical isolates from around the world [86]. As expected, not much variation was detected between NF54 and the 3D7 clone derived from it, but thousands of SNPs, indels and small structural variations, many of which fall in immunologically important regions, were identified in comparisons with the heterologous strains. These results make a clear case for the profound impact of parasite diversity on vaccine development.

## 4. The Disease: The Merozoite

Once the merozoites transition from the liver to the bloodstream, they infect erythrocytes, initiating the blood cycle. This is a complex process involving many parasite proteins and host cell receptors interactions that are highly species-specific. *P. falciparum* uses several invasion pathways, some of which are redundant and interchangeable [87,88]. The first step is the attachment to the erythrocyte through merozoite surface proteins (MSPs). MSP1 is a major player in initial attachment, forming a complex with merozoite surface protein duffy binding ligands (MSPDBL1 and MSPDBL2) and mediating interactions with erythrocyte proteoglycans and membrane proteins [89]. Other proteins of this family, MSP3, MSP6 and MSP7, are also part of this complex, and MSP2 and MSP4 seem to be essential for invasion as well [90]. The merozoite rotates until the apical end contacts the erythrocyte membrane, secreting invasion ligands from the apical organelles, the rhoptries and micronemes. These are erythrocyte binding antigens (EBA-175, EBA-140, EBA181 and EBL1) and reticulocyte binding-like homologues (Rh2a, Rh2b, Rh4 and Rh5) that interact specifically with erythrocyte receptors [91]. The secretion of apical membrane antigen 1 (AMA1) from the micronemes forms a tight-junction together with the rhoptry neck protein 2 RON2, initiating internalisation of the merozoite powered by an actin–myosin motor that pulls the erythrocyte membrane around the parasite [91]. During this process, parasite surface proteins including AMA1 and MSP1 are cleaved by proteases and shed into the erythrocyte and the blood stream. Then the parasite grows to trophozoite and schizont, which will burst out of the erythrocyte, releasing merozoites which will continue the infection.

Targeting the blood cycle for vaccine development is attractive because this is the stage that causes most of the clinical symptoms of the disease (Figure 3). It is also when merozoites are periodically released into the bloodstream and therefore briefly free and accessible to the immune system and unsuccessful merozoites linger for even longer periods of time. Furthermore, egress and invasion result in the release of parasite proteins that can strengthen the immune response. The proteins involved in erythrocyte invasion are ideal targets for vaccine development because antibodies against them have been shown to interfere with this process [90,92]. Interference with the blood cycle would prevent disease and potentially also transmission.

The invasion protein **AMA1** is a leading target for vaccine development because of its high immunogenicity and the ability of antibodies against it to block invasion [93]. An AMA1 vaccine was trialled in Malian children [31] resulting in similar infection rates in the vaccinated and non-vaccinated groups (48.4% and 54.4%, respectively), for an efficacy of 17%. A higher level of protection was reported for cases of infection with a parasite homologous to the vaccine strain (64%), but both waned with time to 24% with homologous and 10% heterologous parasites in a 24 month follow-up study [32]. Disappointingly, the AMA1 vaccine showed no impact on the rate of experimental infections in controlled infection trials with the homologous strain 3D7 [33,34]. AMA1 is highly polymorphic; the genetic diversity includes amino acid changes in regions involved in invasion [66,94], and the extensive variation has been described even at the regional level [95,96]. This protein was combined with MSP1 in an attempt to increase efficacy, but protection levels against CHMI with the homologous parasite remained low (11%) [35], suggesting that antigenic polymorphism cannot be overcome by multiepitope designs.

To overcome AMA1 polymorphism, a diversity-covering formulation, **AMA1-DiCo,** was designed incorporating three variants to represent the major haplotypes based on 355 sequences available at the time, to provide broader protection [97]. A phase Ia/Ib trial was conducted with French and Burkinabe volunteers that reported the induction of antibodies reactive to parasites from different strains [98]. A combination with its natural partner in the invasion process, PfRon2, in an attempt to increase the potency of the IgG antibodies induced has also been tried [99]. However, though this combination was shown to improve the protection conferred by AMA1 alone to *P. falciparum* infection in *Aotus* monkeys [100], Ron2 did not improve cross-strain antibodies in humans, nor in vitro growth inhibition [99]. These data suggest it would be very difficult to cover the worldwide haplotypes in order to confer a global protection against malaria.

The merozoite protein **MSP3** was identified as a vaccine candidate by the association of antibodies against it with parasite growth inhibition and protection in passive antibody transfer experiments [101]. While MSP3 is a highly variable protein, the C-terminus has been shown to be relatively conserved among parasite isolates [102,103]. A vaccine was developed with the C-terminus of MSP3 from *P. falciparum* Fc27 or 3D7, strains and several clinical trials have showed the induction of specific antibodies and cellular immunity [104,105,106,107]. Naturally developed antibodies against MSP3 are associated with a reduced risk of infection [108], but allele-specific antibodies showed a stronger correlation with protection than antibodies to the relatively conserved C-terminus. A diversity study in Thailand found two major haplotypes, designated 3D7 and K1, whose contribution to an effective immune response argues in favour of the inclusion of these sequences in a vaccine formulation. However, expanding the analysis to a wider collection of isolates from around the world painted a much more complex picture of the prevalent haplotypes in different areas [109], indicating that the two major haplotypes K1 and 3D7 are unlikely to cover the global variation in this protein.

In a renewed effort, the C-terminus of MSP3 was fused with a region of the glutamate-rich protein GLURP in the **GMZ2** vaccine. GLURP is expressed through the pre-erythrocytic and erythrocytic stages, and has three major regions (GLURP_94–489_ (R0), GLURP_489–705_ (R1), and GLURP_705–1178_ (R2)) that elicit IgG responses. A strong correlation between IgG anti R0 and R2 with protection against malaria infection was reported [110]. However, a phase II trial in various countries in Africa showed an efficacy of 14% and 27% against severe malaria [36], while a CHMI trial with a homologous parasite failed to confer any protection against a PfSPZ challenge [37]. The low efficacy is likely to be related to the high variability of GLURP, whose only region of relative conservation is R2 [66].

Other more conserved surface proteins are being tested for their capacity to elicit a protective immune response—the trophozoite exported protein TEX1 and the serine repeat antigen 5 (SERA5) [66]. TEX1 is exported to Maurer’s Clefts in the erythrocyte and its function has not yet been determined. A highly conserved segment from **TEX1**, **Pf27**, that corresponds to a sequence predicted to assume a random coiled coil of 104 aminoacids (P27A), was found to be highly immunogenic and have growth inhibitory capacity in assays in vitro [111]. The genetic diversity of this peptide was limited to three non-synonymous SNPs, of which two were present in only one isolate examined worldwide [112]. There are great expectations for further trials with this peptide.

**SERA5** is an abundant blood stage antigen present in the lumen of the parasitophorous vacuole and released into the blood stream following schizont egress. It was selected for clinical development on the basis of epidemiological studies showing high antibody titres that inversely correlated with malaria symptoms and severe disease [113]. In vitro studies of parasite growth inhibition by specific antibodies and animal studies demonstrated protection against *P. falciparum* challenge in non-human primates [114]. A recombinant form of the SERA5 N-terminal domain, SE36, based on the *P. falciparum* Honduras-1 strain showed significant protective efficacies only for more severe diseases, suggesting that BK-SE36 could have a disease-ameliorating rather than preventive effect [38]. The analysis of 445 near full-length *P. falciparum* SERA5 sequences from nine countries in Africa, Southeast Asia, Oceania and South America revealed extensive variations in the number of octamer repeat (OR) and serine repeat (SR) regions, as well as substantial levels of single nucleotide polymorphism (SNP) in non-repeat regions. Remarkably, a 14 amino acid sequence of SERA5 (amino acids 59-72) that is known to be the in vitro target of parasite growth inhibitory antibodies was found to be perfectly conserved in all 445 worldwide isolates of *P. falciparum* evaluated [115].

More recently, **Rh5** emerged as the essential ligand for the *P. falciparum* invasion of erythrocytes [116], which makes it an ideal vaccine candidate that circumvents all the redundant alternative invasion pathways the parasite has devised through evolution. Rh5 has been found to be relatively conserved in isolates from several countries [117], and also more globally, with most of the diversity localised to the central region of the protein [66]. A full length Rh5 vaccine from the *P. falciparum* 3D7 strain induced antibodies in mice that were capable of inhibiting worldwide strains of the parasite in vitro [118]. A phase Ia clinical trial showed that an Rh5 vaccine induces antibodies with cross-strain in vitro growth inhibition activity [119]. Further trials to demonstrate the efficacy of the vaccine are ongoing, and given the essentiality of this protein, there is great interest in the outcome.

## 5. Whole Asexual Parasites Vaccine

With the aim of maximising antigen presentation to the immune system, a vaccine consisting of blood stage chemically inactivated asexual parasites was designed. The whole parasite approach is intended to induce immunity to a high number of parasite antigens, including unknown ones together with a mixture of conserved and variable proteins. In vivo studies in *Aotus* monkeys showed that the chemical inactivation of the FVO *P. falciparum* parasite is effective, and that vaccination induces parasite-specific T-cells, but not antibodies [120]. In the first clinical study of this kind, malaria-naïve participants received one dose of chemically inactivated 7G8 *P. falciparum* parasites [121]. The results showed a variable IgM response, lymphocytic proliferation induced by homologous as well as heterologous parasites, and an increase in various cytokines, including IFN-gamma. There are no data yet about the parasiticidal and protective capacity of the immune response induced by this vaccine. It will be crucial to test the cross-strain protection levels of a whole parasite vaccine to evaluate if the inclusion of all antigenic proteins can overcome the high genetic diversity of worldwide *P. falciparum* strains.

## 6. Transmission: The Gametocytes

Gametocytes differentiate in the blood when infection is high or the health of the human host deteriorates, putting in jeopardy the survival of the parasite (Figure 4). Female and male stages are taken by the mosquito during the blood meal, which mate in the mosquito to form a zygote, and it is at this point that genetic recombination can happen [122]. The development in the mosquito culminates with the formation of sporozoites that migrate to the salivary glands and are deposited in the next human host during the mosquito’s next blood meal [123]. Vaccines targeting the gametocyte stage aim at reducing or stopping the transmission of malaria from one person to the next, rather than preventing clinical disease in any one person. Interruption of transmission has been proven many times throughout the history of the battle with malaria to be very effective at reducing the overall burden of the disease; these include the fumigations in 40 s–60 s to reduce mosquito populations, and more recently the introduction of insecticide-impregnated bednets to prevent mosquito bites. However, these initiatives proved to have complications, such as the toxicity of the insecticides, the resistance of the mosquitoes to them and changes in mosquito behaviour to diurnal habits. Targeting the gametocytes is one potential avenue to stop transmission and lower the burden of malaria in communities.

A variety of proteins with different patterns of expression have been used to induce a transmission-blocking immune response. The antigens Pfs230C and Pfs48/45 are expressed in gametocytes within the human host, while Pfs25 and Pfs28 are expressed in the late gametocyte, zygote and oocyst stages of the parasite in the mosquito midgut. Some novel proteins have also been considered, including the male gametocyte HAP2, the female gametocyte Pfs47 and the mosquito APN1 [124].

Of these proteins, vaccines prepared with NF54 sequences for **Pfs25** and **Pfs230C** conferred a complete blockage of NF54 *P. falciparum* parasites in *A. stephensi* mosquitoes [125], reducing oocyst counts in an antibody concentration-dependent manner. Tests for efficacy against heterologous parasites were also performed using isolates from children in Burkina Faso. The efficacy of Pfs25 in inducing transmission-blocking antibodies with activity against heterologous parasites was confirmed with *P. falciparum* isolates from Thailand [126]. The sequencing of Pfs25 among isolates revealed that this protein is highly conserved. Wider sequencing studies of 329 isolates from various countries in Africa and Asia confirmed the conservation of Pfs25, detecting only 1 polymorphism in the region associated with the binding of transmission-blocking antibodies [127]. This was in contrast with Pfs48/45, which showed eight non-synonymous mutations, four of which affected transmission-blocking epitopes. Though Pfs230C also demonstrated complete transmission blocking activity, it was found to be much more diverse than Pfs25 [4], so heterologous activity with a wider selection of geographical isolates would need to be established. The high conservation of Pfs25 and its effectiveness in inducing transmission-blocking antibodies show potential for this strategy to control malaria.

## 7. Multi-Stage Vaccine

A widely tested synthetic multiepitope and multistage vaccine against *P. falciparum* was developed in Colombia over 30 years ago. The synthetic *P. falciparum* vaccine 66 (**SPf66**) consisted of peptides derived from the Colombian FCB-2 strain, including three merozoite proteins: MSP1, SERA, and an unidentified molecule and the NANP repeat from the sporozoite CSP [128]. The vaccine was trialled in Colombia [129] and neighbouring countries Venezuela [130] and Ecuador [131], with a reported efficacy of the vaccine ranging between 35 and 60%. However, when tested further afield in the Rondonia state in Brazil [132], the efficacy dropped to 14% for the first episode of malaria. When testing was expanded to different continents, the drop in efficacy went from 31% in Tanzania [133] to 8% in Gambia [134], and no protection was reported in Thailand [135]. Though all these trials were set up and evaluated differently, most of them reported the induction of an immune response to the vaccine, which makes a compelling case for the role of parasite genetic diversity in the drop in vaccine efficacy in different geographic regions.

Another vaccine targeting various stages of the parasite was **NYVAC-Pf7**, composed of seven antigens derived from the *P. falciparum* 3D7 strain including three pre-erythrocytic (CSP, TRAP and LSA-1), three blood stage (MSP1, SERA and AMA1) and the Pfs25 sexual stage antigen [136]. This vaccine was tested in Phase I/IIa trials showing high levels of cellular immunity even if the antibody response was overall weak [137]. CHMI with the homologous 3D7 strain resulted in 1/35 sterile protection and a significant reduction in the time to parasitemia. More recently, a combination of AMA1 and CSP in the NMRC-M3V-Ad-PfCA vaccine showed no [39] or modest protection [40]. A more complex combination, containing constructs with AMA1-DiCo and Pfs25 together with either CSP, MSP1 or Rh5, was used in the immunisation of rabbits [138]. Efficacy assays with rabbit antibodies showed promise, but evidenced a need to improve the induction of reactive antibodies. Potentially, the use of AMA-DiCo would give these formulations a broader coverage, but as noted above, it is unlikely to be worldwide.

## 8. Lessons for the Future

The ideal vaccine should induce universal, long-term protection with the fewest possible doses. The current vaccines discussed here, as well as many previous attempts and trials, have been developed from *P. falciparum* lab strains. In most cases, the efficacy was promising for homologous strains but declined significantly with increasingly geographically distant heterologous parasites. Single-antigen vaccines stimulate an immune response of variable strength and that is generally strain-specific. Multi-stage vaccines improve the potency of the immune response, but not the cross-reactivity. Whole parasite strategies elicit strong and broader immunity, including conserved epitopes, but though sterile protection can be achieved with homologous strains, protection against heterologous parasites drops significantly. Designs covering some of the diversity of specific antigens show a broader antibody response, but generally also highlight the difficulty in achieving overall coverage for a universal vaccine.

We have advanced a long way from the initial anti-*Plasmodium falciparum* malaria vaccines, which viewed the species as one parasite. Antigen diversity has started being increasingly appreciated, particularly since the *P. falciparum* genomic sequence has been completed [139], but the full extent of genomic variation was only unveiled by the vast sequencing of worldwide parasite samples in the last decade. With these data, there is now a much more complete picture of the genomic characteristics of parasites in different regions, the prevalence of multiclonal infections, and the dynamics of parasite populations that will necessarily have an impact on the efficacy of vaccine efforts. The availability of these vast sets of genomic data and the advent of informatics capabilities must be taken full advantage of in order to guide the identification of vaccine candidates and design new approaches. The existing resources allow an in-depth analysis of genetic diversity and the categorising of the world haplotypes into groups with common features. The sequence analysis can also be complemented with data from immunological studies to select the most immunogenic and conserved peptides.

Perhaps a design including peptides from several antigens, each one covering the major haplotypes, would increase strain coverage, since the haplotypes for each peptide are unlikely to overlap the same strains. A synthetic vaccine of this kind gives more flexibility than a whole parasite approach, where the number of strains that can be included is limited. However, gene editing technologies, guided by genomic data, could perhaps be exploited to introduce the most prominent haplotypes of different antigens into one engineered ‘universal parasite’. Alternatively, it might become more feasible to consider regional vaccines covering specifically local haplotypes.

The latest data and technology will be useful to identify specific epitopes that are conserved enough to be used in a vaccine either on their own or in a combination of haplotypes to trigger an efficient immune response. This needs to be accompanied by the development of novel tests to measure and assess the efficacy of the new generation vaccines to bring them as close to humans as possible before deploying clinical trials. Some of the recent efforts are already using these advances to improve the chances of defeating this parasite.

## Figures and Tables

**Figure 1 microorganisms-08-01748-f001:**
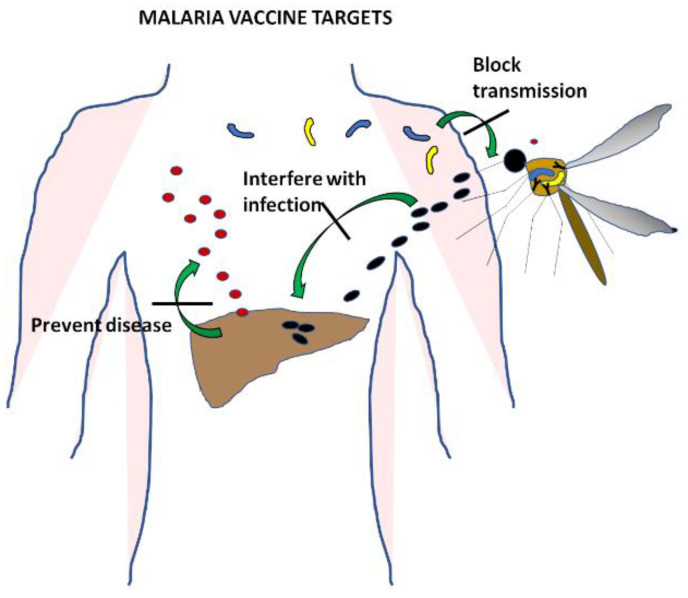
Malaria parasite life cycle. Stages of malaria infection that are the focus of vaccine development. Interfering with the infection of hepatocytes would prevent the disease altogether. Blocking the blood stage would stop clinical symptoms and also prevent transmission of the disease. Targeting gametocytes of early stages in the mosquito would avoid infection spread from person to person without affecting the development of clinical disease.

**Figure 2 microorganisms-08-01748-f002:**
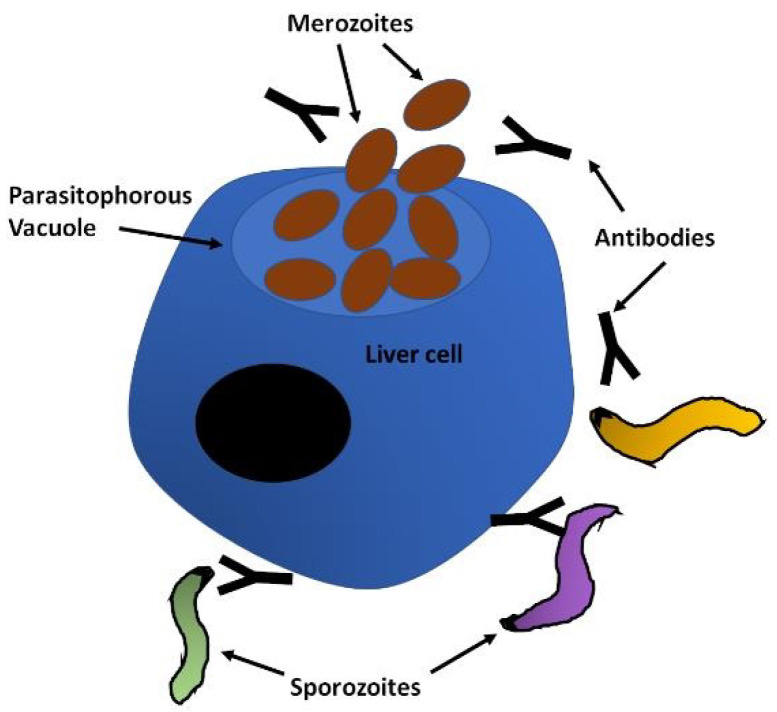
The liver stage. Sporozoites leave the skin and migrate to the liver, where they invade hepatocytes. Surface proteins of the sporozoite are involved in the migration, recognition and invasion of hepatocytes. These are represented by the dense black outline of the sporozoites and increased density in the apical region, and include mainly CSP, TRAP and AMA-1 at this stage. After the intracellular replication of the parasite the first merozoites that will initiate the blood infection are released into the blood stream. Pre-erythrocytic vaccines aim to induce an immune response, humoral and cellular, to interfere with invasion and trigger elimination of the parasite, respectively. CSP (circumsporozoite protein); TRAP (thrombospondin-related anonymous protein); AMA-1 (apical membrane antigen 1).

**Figure 3 microorganisms-08-01748-f003:**
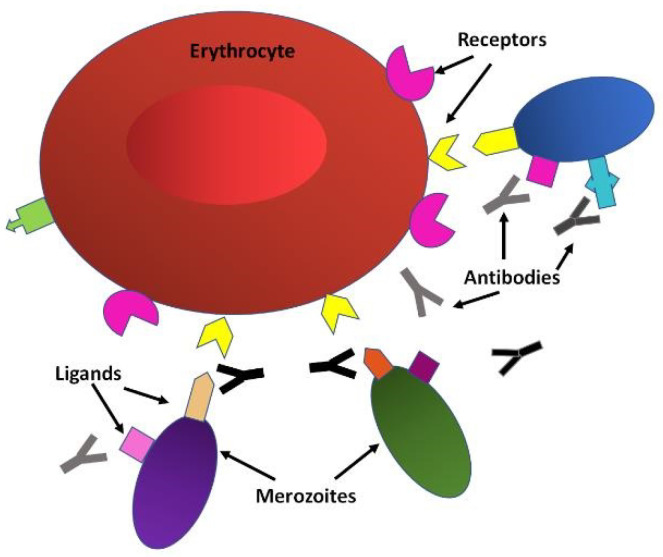
The blood cycle. Merozoites with different haplotypes (represented by different colours) of surface ligands (represented by different shapes) bind to erythrocyte receptors to start the invasion process. Erythrocytic stage vaccines aim to induce antibodies that recognise the parasite’s ligands to interfere with invasion and mediate the elimination of the parasite.

**Figure 4 microorganisms-08-01748-f004:**
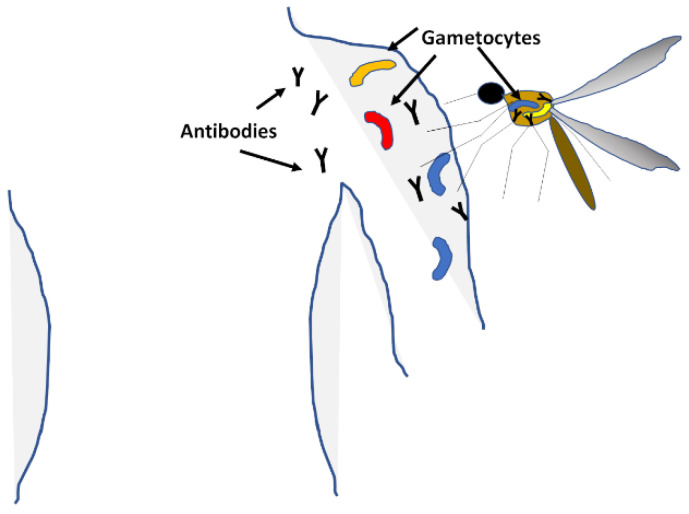
**Transmission.** Male and female gametocytes form in the blood stream and migrate to the skin, where they are taken up by feeding mosquitoes. Transmission-blocking vaccines aim to mount an immune response against the gametocytes or the early development stages in the mosquito to eliminate the sexual stages and reduce the oocyte load in the vector.

**Table 1 microorganisms-08-01748-t001:** Recent clinical trials of Plasmodium falciparum vaccines with efficacy data available. Orange: pre-erythrocytic proteins; red: whole sporozoite vaccines; green: blood stage proteins; blue: multi-stage vaccines.

Antigen	Vaccine	Strain	Homologous	Heterologous	Trial	Reference
**Multi Epitope TRAP**	ME-TRAP	T9/96		21%	Phase I/Iia: NCT00890760	[9]
				13%	Phase I/Iia CHMI NCT01623557	[10]
				13%	CHMI	[11]
				67%	Phase Iib: NCT1666925	[12]
				10.30%	ISRCTN05221133	[13]
				No protection	Phase I/IIb: NCT01635647	[14]
**CSP**	RTS,S/AS01	3D7		36.30%		[15]
				34.80%	Phase III: NCT00866619	[16]
				30.10%		[17]
				27%		[18]
				32.1–53.7%	Phase III: NCT02207816	[19]
			86.70%		CHMI Phase Iia: NCT01857869	[20]
			55–76%		Phase IIa: NCT03143218	[21]
**Me-TRAP+RTS,S**			82.40%		CHMI NCT01883609	[22]
**Whole Sporozoite**	PfSPZ	NF54	100%		CHMI NCT01441167	[23]
			100%		CHMI NCT00442377	[24]
			100%		CHMI NCT02613520	[25]
				29%	Phase I: NCT01988636	[26]
				20%	CHMI NCT02132299	[27]
			92.30%	80%	CHMI NCT02215707	[28]
			100%		CHMI NCT02115516	[29]
			100%	11–20%	CHMI NCT02098590	[30]
**AMA1**	FMP2.1	3D7	64.30%	20%	Phase II: NCT00460525	[31]
			24%	9.90%	Phase II: NCT00460525	[32]
			0%		CHMI NCT02044198	[33]
			0%		CHMI NCT00385047	[34]
**AMA1+MSP1**	ChAd63-MVA	3D7	11%		CHMI NCT01142765	[35]
**MSP3- GLURP**	GMZ2	FVO / F32		14%	PACTR2010060002033537	[36]
				0%	CHMI PACTR201503001038304	[37]
**SERA5**	BK-SE36	Honduras-1		75%	Phase Ib: ISRCTN71619711	[38]
**CSP+AMA1**	NMRC-M3V-Ad-PfCA	3D7	0%		Phase I/IIa: NCT00392015	[39]
			27%		CHMI NCT00870987	[40]

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
