# Peer review of "Diversify and Conquer: The Vaccine Escapism of Plasmodium falciparum"

_microorganisms, 2020, doi:10.3390/microorganisms8111748_

Round 1
Reviewer 1 Report
This is a comprehensive review of the myriad P. Falciparum vaccine candidates currently under consideration. This is a very timely topic and therefore also a very common review topic. However, I feel that this particular review is unique enough in emphasis that it provides new information to the malaria community. The review focuses on the diversity of the vaccine candidates and the role of this diversity in leading to the poor efficacy of many of the vaccine candidates trialed to date. This diversity leading to vaccine ‘failure’ is not a new concept – but this review very extensively brings together the diversity data for all of the antigens currently under trial. It will therefore be a nice resource and I suspect will be consulted and referenced often.
There are several VERY minor comments.
I suspect that the PMID in the last row of Table 1 is incorrect
Line 138 – typo on 3D7?
Figure 3 – What is the green ‘receptor’ in the upper right?? Was this meant to be a receptor for the merozoites? All the merozoites seem to have ‘antigens’ that look the same (which is inaccurate). The Mz antigens might be changed to have different shapes to match the complimentary receptors on the RBC. Also (very small point) – what is the two tone of the RBC supposed to represent? Is the lighter tone in the middle supposed to represent the bi-concave disc shape? It might be mis-construed to be a nucleus. Perhaps just a homogenous color (or more subtle shading to represent the bi-concavity) would be more appropriate.
Line 264 – weaned vs waned
Line 420 “unpick specific epitopes”. I am not sure that I understand this phrase – or this sentence.
Author Response
Dear reviewer,
Thank you for taking the time to read this manuscript and for your helpful comments.
The PMID of the last article of table I has been added
3D7 was corrected in 138
Figure 3 has been improved. the bi-concavity representation has been made a bit darker to make it more subtle so that it is clear that this corresponds to the shame of the cell rather than a nucleus. The labelling of the erythrocyte receptors has been improved and a labelling of the ligands has been added. The legend of this figure has also been revised including an explanation of the receptors (different shapes) and their haplotypes (different colours), to make the representation clearer.
Waned has been corrected in ine 246
Line 420 has been re-written to clarify the meaning: The latest data and technology will be useful to identify specific epitopes that are conserved enough to be used in a vaccine either on their own or in a combination of haplotypes to trigger an efficient immune response.
Many thanks and best wishes,
Alena Pance
Reviewer 2 Report
The review by Alena Pance et al discuss different vaccines against Plasmodium falciparum. The review does a good job in discussing the strength and limitations of various approaches.
- In Table Reference should not be in PMIDs.
- Senetence not clear “ In this genomic era, the complete sequencing of the Plasmodium falciparum genome [3] uncovered the full set of genes paving the way to start deciphering their function in the different stages of the life cycle.”
- “Importantly, these studies are revealing the complexity of parasite diversity, providing 47 a detailed insight into the genetic variation of parasite proteins at the centre of vaccine development.” How parasite proteins variation at the Centre of vaccine development?
- Figure 2 should show the parasite antigens such as CSP, TRAP and AMA-1.
- “study found an efficacy of 10% [32] and in Kenyan children no significant difference between the control and vaccinated groups was observed [33].” Sentence is unclear.
- Author should provide image of TRAP and CSP domains used for vaccines and immunogenic epitopes.
- “ntinents [54], which can be also observed in PlasmoView” What is Plasmoview ? Please provide details.
- “The assessment of the efficacy of this vaccine will prove very interesting, particularly in the context of heterologous infections.” What is interesting here is not clear.
- “Disappointingly, controlled infection trials with the same 3D7 strain of the AMA1 vaccine did not find any impact of the vaccine on the rate of experimental infections” Please improve grammar.
- Author should provide some background on TEX1, Serine Repeat Antigen 5 (SERA5) and
Author Response
Dear reviewer,
Please find below details of the changes introduced in the manuscript following your comments:
1- The PMIDs in table I have been replaced with the corresponding references in the text.
2- This sentence was modified to: In this genomic era, the completion of the whole sequence of the Plasmodium falciparum genome [3] revealed the full set of genes, paving the way to start deciphering their function in the different stages of the life cycle of the parasite.
3- This sentence has been changed to: Importantly, these studies are revealing the complexity of parasite diversity, providing a detailed insight into the genetic variation of parasite proteins that are at the centre of vaccine development.
4- Figure 2 has been changed to depict the involvement of surface proteins in the invasion of hepatocytes, while maintaing consistency with the rest of the figures of the manuscript. These features have been clarified in the legend.
5- This sentence was changed to: Natural infections after vaccination of adult volunteers reduced the risk of infection by 67% [31]. Another study found an efficacy of 10% in Gambia [32], while in Kenyan children no significant difference between the control and vaccinated groups was observed [33].
6- Given that CSP and TRAP are some of the major vaccine candidates, there are myriads of images and diagrams available depicting in great detail these proteins as well as the domains used in the vaccines. As this manuscript does not focus solely on these vaccines but rather gives an overview of the parasite entigens that have been used in the development of vaccines, the contribution of such image to the arguments proposed is unclear. In order to preserve consistency throughout the manuscript and the sense of a broader discussion, an extra figure of these antigens in particular was not deemed appropriate.
7- Plasmoview is now clearly explained when first cited in the manuscript, line 113/114 and a reference to guide readers to it is provided: 40.
8- This sentence refers to the assessment of the efficacy of this vaccine because the results of the trials have not yet been published. As the immune epitopes seem to be localised to fairly conserved regions of the protein, it will be very important to determine whether this vaccine can provide protection, particularly to heterologous parasite strains.
9- This sentence was changed to: Disappointingly, the AMA1 vaccine showed no impact on the rate of experimental infections in controlled infection trials with the homologous strain 3D7
10- This existing information on TEX1 has been added: TEX1 is exported to Maurer’s Clefts in the erythrocyte and its function has not yet been determined.
Background about SERA5 has also been added: SERA5 is an abundant blood-stage antigen present in the lumen of the parasitophorous vacuole and released into the blood stream following schizont egress.
Best wishes,
Alena Pance
Reviewer 3 Report
Useful mini-review/position paper. Nicely presented. Some very minor editing suggestions:
Line 59: remove extra period
Line 69: address positioning of the citations for reference 14 and 15 (remove period and combine)
Line 73: italicize Plasmodium
Line 90: CSP has previously been defined (line 64)
Line 260: Aotus should be italicized
Line 288: Is Pf27 intentionally bold?
Line 319: Aotus should be italicized
Font differences in Section 8 should be addressed
Author Response
Dear reviewer,
Thank you for taking the time to read this manuscript and for your helpful comments.
The extra period in line 59 has been removed.
References 14 and 15 have been correctly formatted.
Plasmodium in line 73 has been changed to italics
Circumsporozoite protein in line 90 has been deleted
Aotus in line 260 has been changed to italics
Pf27 in line 288 is intentionally bold, to highlight this particular peptide as a vaccine candidate.
Aotus in line 319 has been changed to italics
The formatting of section 8 has been corrected
Many thanks and best wishes,
Alena Pance
This manuscript is a resubmission of an earlier submission. The following is a list of the peer review reports and author responses from that submission.
Round 1
Reviewer 1 Report
In the manuscript entitled "Diversify and conquer:..." the authors describe the different approaches going on in regard to a vaccine against malaria. The manuscript is well written and comprehensive.
In regard to the manuscript entitled "Diversify and Conquer: The tribulations of a Malaria Vaccine", is my opinion that the manuscript is well written. The author reviews the research and development of the vaccine, giving a nice and easy to follow historic perspective as well as the state of the issue. In the manuscript the author included the research that span over 50 years. In regard to the science, the authors revision of the topics seems appropriate. Since it is a review there is no new information, just interpretation and summary of findings in the development of the malaria vaccine.Reviewer 2 Report
Microorganisms_review June 23, 2020
The review paper presented, is written with a very broad overview of the subject. The author went to great length to amass the information. Greater story-telling style could be adopted.
Firstly, author is encouraged to define the review manuscript, as well as how the articles were amassed. If this is a scoping review then it should be stated, vs. systematic review, vs. comprehensive (www.joannabriggs.org) and a materials and methods section written for keyword search, protocol registered with JBI, and PRISMA flow chart included.
Secondly, it is my impression that there are two distinct writing styles in the manuscript that could have benefited from further review and editing (e.g. 221-245 vs. 365-390).
Some specific comments:
When searching PubMed “malaria vaccines” yielded >1500 review articles. It would be helpful to the reader to define at the end of the introduction what content this review intends to cover, specifically, and how this review adds or differentiates from other currently written review.
Acronyms should be spelled out once and then formatted accordingly throughout the text (e.g. TRAP is listed on line 45, but only spelled-out on line 60, and again on line 78). Please revise all, and throughout the entire text. Consider including an abbreviations section.
According to APA-style number <10 should be spelled-out, re. line 53, line 329.
Line 37, when stated “flow” is this to mean “spread”.
Generally, the paragraphs, could benefit from more structure, either as a historical (chronological) perspective, or according to levels of success of vaccines…there should be a formal topic sentence, main body of the paragraph, and then a concluding sentence. Not all of the paragraphs are structured the same way. Some include animal studies, others do not…there is a lack of consistency.
Try not to start sentences with “but”.
In vitro should be italicized on line 203.
There is text bolded, line 64, 78, 85, and others…was this intentional?
Could some of these sections have been summarized as tables, i.e. line 154-168; 180-196?
Section 8. needs considerable attention, it is currently riddled with run-on sentences, some issues with subject-verb agreement. The entire section needs to be re-written (see attached).
The manuscript does not appear to have undergone a peer-review editing process prior to submission to Microorganisms. Should be read by at least x2 peers, one w/ content expertise, one with limited, or little.
Sentences that needs revision/re-read, possible run-on sentences: Line 53-55; 59-61; 64-66; 91-93; 105-113; 119-122; 129-130; 136-139 (consider adding an image of main antigens of interest); 146-148; 154-168; 186-192; 198-200; 210-211.
New paragraphs recommended on line 97, 132.
Should sub-categories be listed such as at line 169.
Line 188 should amino acid, two separate words.
Line 333, is the a way to organize/distinguish the animal work better, either as a separate table, or as a separate paragraph for each section.
Line 187-188, a table would have been good, with 1st author, PMID, % protection, year, country of origin, homo vs. heterologous, as mentioned above.
Figures were not included with document. Was unable to evaluate.

Reviewer 3 Report
This is a review of malaria vaccines, the development of which the author indicated has been much helped by the modern techniques of genetic analysis and gene-based technology. The manuscript lists the various protein components of the parasite and the attempts over the years to use these components as vaccines. Most of these are straightforward and just presented in an insipid manner. The whole manuscript is just a pedantic accumulation of knowledge extracted from the published literature database. It is not clear what exactly the goal of this review was. The introduction seemed to suggest that parasite variations are going to be discussed and how parasite variations or isolate variations have thwarted the development of a global malaria vaccine. However, the title, the introduction and the body of the article are not tied together at all. There is no novelty of thought for the data of the analysis in this manuscript.
The writing is very diffuse. The language is colloquial, and it is not scientific nor crisp and lacks precision. The information has not been analyzed and presented clearly. This is a superficial and weak manuscript and cannot be recommended for publication in a high-quality scientific journal.
The literature search has gaping holes in the malaria vaccine literature. A very selective (perhaps biased) selection of references has been cited. Therefore, this review paper is lacking all the information on malaria vaccines even if only focused on P. falciparum.
This review is not going to add to the knowledge for the field, nor does it possess an edge over other vaccine related review.
As examples, following critiques are provided.
INTRODUCTION
- Vaccine efforts have also been expended on vivax over the years. This also should be discussed and referenced.
- Lines 42-49 – Very poorly written and diffuse. The infection process is loosely described, and it will be clearer if authors specified when which epithelia and endothelia sporozoites need to traverse at each stage. A layperson (a scientist not familiar with malaria will have a difficult time understanding the infection process)
- Abbreviations and/or acronyms must be stated in full at the first appearance.
- Is the manuscript focused only on a vaccine for plasmodium falciparum? Or is it a general review on Malaria vaccines - it is not clear.
LESSONS FOR THE UTURE
- Line 354 - anti plasmodium falciparum malaria vaccine is an awkward phrase and is redundant.
- “ …decreasing efficacy when trialled further afield.” This phrase is too specific to a region Put an international audience please use a different way to describe.
- This section is very diffuse and not quite clear what the lessons learned were. What the author is suggesting we need to do in clear terms would be extremely helpful. The language is very nonscientific.